# PIECEWISE LINEAR NEURAL NETWORK VERIFICATION: A COMPARATIVE STUDY

## ABSTRACT

The success of Deep Learning and its potential use in many important safety-critical applications has motivated research on formal verification of Neural Network (NN) models. Despite the reputation of learned NN models to behave as black boxes and the theoretical hardness of proving their properties, researchers have been successful in verifying some classes of models by exploiting their piecewise linear structure. Unfortunately, most of these approaches test their algorithms without comparison with other approaches. As a result, the pros and cons of the different algorithms are not well understood. Motivated by the need to accelerate progress in this very important area, we investigate the trade-offs of a number of different approaches based on Mixed Integer Programming, Satisfiability Modulo Theory, as well as a novel method based on the Branch-and-Bound framework. We also propose a new data set of benchmarks, in addition to a collection of previously released testcases that can be used to compare existing methods. Our analysis not only allows a comparison to be made between different strategies, the comparison of results from different solvers also revealed implementation bugs in published methods. We expect that the availability of our benchmark and the analysis of the different approaches will allow researchers to develop and evaluate promising approaches for making progress on this important topic.

## 1 INTRODUCTION

Despite their success in a wide variety of applications, Deep Neural Networks have seen limited adoption in safety-critical settings. The main explanation for this lies in their reputation for being black-boxes whose behaviour can not be predicted. Current approaches to evaluate the quality of trained models mostly rely on testing using held-out data sets. However, as Edsger W. Dijkstra said (Buxton & Randell, 1970), "testing shows the presence, not the absence of bugs". If deep learning models are to be deployed to applications such as autonomous driving cars, we need to be able to enforce and verify safety-critical behaviours.

A particularly illustrative instance of the limit of our understanding of the behaviour of learned models lies in the existence of adversarial examples (Szegedy et al., 2014): small perturbations, imperceptible to the human eye, are capable of significantly modifying the predictions generated by a network, despite it performing well on its test set. Several methods have been proposed (Gu & Rigazio, 2015; Goodfellow et al., 2015) to make networks more robust to those perturbed inputs. It is however not clear if those methods are successful at reducing the number of adversarial examples or if they are just capable of reducing the number of adversarial examples that currently known methods can generate. The only way to know this is to measure the size of the region around training samples guaranteed to not contain adversarial examples.

Some researchers have indeed tried to use formal methods to obtain guarantees like the one mentioned above. To the best of our knowledge, Zakrzewski (2001) was the first to propose a method to verify simple, one hidden layer neural networks, but only recently (Katz et al., 2017a; Narodytska et al., 2017; Cheng et al., 2017c) were researchers able to work with non-trivial models by taking advantage of the structure of ReLU-based networks. Even then, these works are not scalable to the large networks encountered in most real world problems.

A major roadblock in the area has been the lack of any analysis of the success and failure modes of the proposed approaches. To remedy this problem, we gather a data set of test cases based on existing

literature and parametrically explore the space of possible architectures. We use it to evaluate different published approaches, implementing them ourselves where no public version was available, and generate the first experimental comparison of published tools. In addition, we introduce a general formalism for the problem, showing possible directions for improvement, as well as a new method showing significantly better performance on practical scenarios. Additionally, our comparison also revealed bugs in some publicly available NN verification software, made evident by contradictions in the results of the various methods.

## 1.1 PROBLEM SPECIFICATION

We now specify the problem of formal verification of neural networks. Given a network that implements a function $y = f(x)$, a bounded input domain $\mathcal{C}$ and a property $P$, we want to prove that

$$\mathbf{x} \in \mathcal{C}, \quad \mathbf{y} = f(\mathbf{x}) \implies P(\mathbf{y}). \tag{1}$$

A toy-example of such a problem is given in Figure 1. On the domain $\mathcal{C} = [-2; 2] \times [-2; 2]$, we want to prove that the output $y$ of the one hidden-layer network, always satisfy the property $P(y) \triangleq [y > -5]$. We will use this as a running example to explain the methods used for comparison in our experiments.

In this paper, we are going to focus on Piecewise-Linear Neural Networks (PL-NN), that is, networks for which we can decompose $\mathcal{C}$ into a set of polyhedra $\mathcal{C}_i$ such that $\mathcal{C} = \cup_i \mathcal{C}_i$, and the restriction of $f$ to $\mathcal{C}_i$ is a linear function for all $i$. While this prevents us from including networks that use activation functions such as sigmoid or tanh, PL-NNs allow the use of linear transformations such as fully-connected or convolutional layers, pooling units such as MaxPooling and activation functions such as ReLUs and its various extensions such as Leaky ReLU or PReLU (He et al., 2015). In other words, PL-NNs represent the majority of networks used in practice. Note that layers such as Batch-Normalization also preserve piecewise linearity at test-time.

The properties that we are going to consider are Boolean formulas over linear inequalities. Although the formulas in (1) define the property to be a function of the ouptut $\mathbf{y}$, we have no loss of generality: any property involving more variables could be expressed as a function over the output of a different network, including all additional variables in its output.

As the problem of PL-NN verification has been shown to be NP-complete using a reduction from 3-SAT (Katz et al., 2017a), it is unlikely that any polynomial time algorithm will exist. Therefore, experimental comparison remains the only approach possible to evaluate the relative advantages of different methods, which we propose to do.

## 2 RELATED WORKS

We start by briefly presenting related work that we do not include in our comparison as they are not capable of performing general verification of problem of the form of Problem 1.

Zakrzewski (2001) and Hein & Andriushchenko (2017) propose methods based on the second derivatives of the functions expressed by the networks to guarantee that the output of the network doesn't change too much around points. However, this requires the additional assumption that all layers of the networks are twice differentiable, which PL-NN don't satisfy. At the other end of the spectrum, Narodytska et al. (2017) and Cheng et al. (2017c) use SAT solvers to propose verification methods for the specialised case of Binarized Neural Networks (Hubara et al., 2016). These methods, however, don't translate to PL-NNs.

In addition to exact formal methods, some other approaches were proposed that don't provide complete verification. Pulina & Tacchella (2010) proposed a method for verification of networks with sigmoid activation function. The approximation to the non-linearities means their method can not always return a decision for certain problems. Bastani et al. (2016) studied PL-NN in the context of obtaining robustness guarantees against adversarial examples. For scalability reasons, they added additional assumptions, limiting the domain considered to the set of points sharing the same activation pattern than a reference point. This effectively circumvents all of the non-linearities of the network. Xiang et al. (2017) and Huang et al. (2017) both rely on discretisation and perform layer by layer analysis to obtain guarantees over the output, but Huang et al. (2017) require the user to specify a family of possible changes and Xiang et al. (2017) over-approximate the output space

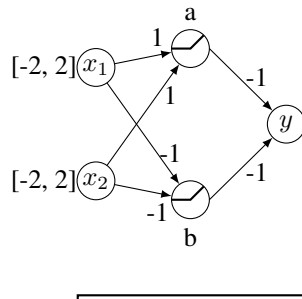

**Prove that** $y > -5$

Figure 1: Example Neural Network. We attempt to prove the property that the network output is always greater than -5

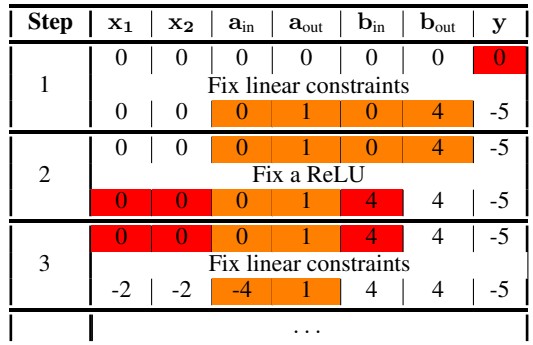

Table 1: Evolution of the Reluplex algorithm. Red cells corresponds to value violating Linear constraints, and orange cells corresponds to value violating ReLU constraints. Resolution of violation of linear constraints are prioritised.

of each layer, leading to the existence of undecidable properties, making both inadequate for our comparison.

## 3 VERIFICATION METHODS

The methods we involve in our comparison all leverage the piecewise-linear structure of PL-NN to make the problem more tractable. We shall use the network of Figure 1 throughout as an illustrative example. All the methods we compare follow the same general principle: given a property to prove, they attempt to discover a counterexample that would make the property false. This is accomplished by defining a set of variables corresponding to the inputs, hidden units and output of the network, as well as the set of constraints that a counterexample would satisfy. For the network of Figure 1, the variables would be $\{x_1, x_2, a_{\text{in}}, a_{\text{out}}, b_{\text{in}}, b_{\text{out}}, y\}$ and the set of constraints would be:

$$
\begin{align}
-2 \leq x_1 \leq 2 \qquad &-2 \leq x_2 \leq 2 \qquad y \leq -5 \tag{2a} \\
a_{\text{in}} = x_1 + x_2 \qquad &b_{\text{in}} = -x_1 - x_2 \qquad y = -a_{\text{out}} - b_{\text{out}} \tag{2b} \\
a_{\text{out}} = \max(a_{\text{in}}, 0) \qquad &b_{\text{out}} = \max(b_{\text{in}}, 0) \tag{2c}
\end{align}
$$

Here, $a_{\text{in}}$ is the input value to hidden unit $a$, while $a_{\text{out}}$ is the value after the ReLU. Any point satisfying all the above constraints would be a counterexample to the property, as it would imply that it is possible to drive the output to -5 or less. However, if this problem is unsatisfiable, no counterexample can exist, which implies that the property is True. We want to emphasize that our requirements go beyond saying that no counterexample could be found: it is necessary to prove that no counter-examples exists. The difficulty stems from the non-linear constraints of (2c). We will now explain how each method tackles this problem.

### 3.1 MIXED INTEGER PROGRAMMING ENCODING

A possible way to eliminate the non-linearities is to encode them with the help of binary variables, which transform the PL-NN verification problem into a Mixed Integer Program (MIP). Lomuscio & Maganti (2017) and Cheng et al. (2017b) advocate the use of big-M encoding to achieve this. For example, the non-linearities of equation (2c) are replaced by

$$
\begin{align}
a_{\text{out}} \geq 0 \qquad\qquad &a_{\text{out}} \geq a_{\text{in}} \\
a_{\text{out}} \leq a_{\text{in}} + (1 - \delta_a)M_a \qquad\qquad &a_{\text{out}} \leq \delta_a M_a \tag{3} \\
\delta_a \in \{0, 1\} \qquad\qquad &
\end{align}
$$

where $M_a$ is a sufficiently large value. The binary variable $\delta_a$ indicates which phase the ReLU is in: if $\delta_a = 0$, the ReLU is blocked and $a_{\text{out}} = 0$, else the ReLU is passing and $a_{\text{out}} = a_{\text{in}}$. The problem remains difficult due to the integrality constraint on $\delta_a$. We provide more details in Appendix A on how to handle MaxPooling units similarly.

This approach has some advantages. As the final problem to solve ends up being a MIP, imposing integrality constraints on the inputs comes at no additional cost. This can prove useful if some input features to the network are known to necessarily be integers (Cheng et al., 2017a). For other methods, imposing these integer constraints would not be possible: either the proof would be attempted

on the relaxed version of the networks or it would have to be done for all possible combinations of integer inputs.

As the solving of MIP is NP-hard, the performance of those methods is going to be dependent both on the quality of the solver used and of the encoding. Cheng et al. (2017b) proposes several methods to obtain a good encoding by picking appropriate values of M, in order for the quality of the linear relaxation of the MIP problem to be as high as possible. However, in the end, the problem is still resolved by a general purpose MIP solver and the question remains open whether a solver not knowing more about the specific structure of the problem can be efficient on challenging benchmarks.

## 3.2 RELUPLEX

Katz et al. (2017a) presents a procedure named Reluplex to verify properties of Neural Network containing linear functions and ReLU activation unit, functioning as an SMT solver using the splitting-on-demand framework (Barrett et al., 2006). The principle of Reluplex is to always maintain an assignment to all of the variables, even if some of the constraints are violated.

Starting from an initial assignment, it attempts to fix some violated constraints at each step. It prioritises fixing linear constraints ((2a) and (2b) in our illustrative example) using a simplex algorithm, even if it leads to violated ReLU constraints (2c). This can be seen in step 1 and 3 of the process shown in Table 1. If no solution to the problem containing only linear constraints exists, this shows that the counterexample search is unsatisfiable. Otherwise, all linear constraints are fixed and Reluplex attempts to fix one violated ReLU at a time, such as in step 2 of Table 1 (fixing the ReLU $b$), potentially leading to newly violated linear constraints. This process is not guaranteed to converge, so to guarantee progress, non-linearities getting fixed too often are split into two cases. This generates two new sub-problems, each involving an additional linear constraint instead of the linear one. The first one solves the problem where $a_{\text{out}} = 0$, the second one where $a_{\text{out}} = a_{\text{in}}$. In the worst setting, the problem will be split completely over all possible combinations of activation patterns, at which point the sub-problems are simple LPs.

## 3.3 PLANET

Ehlers (2017a) also proposed an approach based on splitting the problems over the possible phase of the activation function. Unlike Reluplex, the proposed tool, named Planet, operates by attempting to find an assignment to the phase of the non-linearities. Reusing the notation introduced in Section 3.1, it assigns a value of True or False to each $\delta_a$ variable, verifying at each step the feasibility of the partial assignment. As opposed to Reluplex, this has the advantage of being easily extended to networks containing MaxPooling units.

In order to detect incoherent assignments (such as both $a$ and $b$ being in the greater than zero region of the ReLU in the example of Figure 1) faster, the author proposes a global linear approximation to a neural network. In addition to the existing linear constraints ((2a) and (2b)), the non linear constraints are approximated by sets of linear constraints representing the convex hull of the non-linearities. ReLUs are replaced by the set of equations:

$$a_{\text{out}} \geq 0 \qquad a_{\text{out}} \geq a_{\text{in}} \qquad a_{\text{out}} \leq u_a \frac{a_{\text{in}} - l_a}{u_a - l_a}, \tag{4}$$

where $a_{\text{out}}$ and $a_{\text{in}}$ are respectively the output and input of the ReLU, and $u_{\text{in}}$ and $l_{\text{in}}$ are pre-computed upper and lower bounds of the ReLUs input. This feasible domain is illustrated in Figure 2a. MaxPooling units are replaced by the set of constraints:

$$\forall i, \ \text{out} \geq \text{in}_i \qquad \text{out} \leq \sum_i \left( \text{in}_i - l_{\text{in}_i} \right) + \max_i l_{\text{in}_i}, \tag{5}$$

where $\text{in}_i$ are the inputs to the MaxPooling unit and $l_{\text{in}_i}$ their lower bounds. A one dimensional cut of this is represented in Figure 2b. As a consequence, the whole network is approximated by an LP that can be efficiently queried to detect infeasibility or automatically deducing implied assignments to other $\delta_i$ variables. The values of the lower bounds and upper bounds necessary to define the constraints are built iteratively by optimizing the corresponding variable, based on the constraints imposed by the previous layers. Additional heuristics to make infeasibility detection and implied phase inference faster are described in the original paper (Ehlers, 2017a).

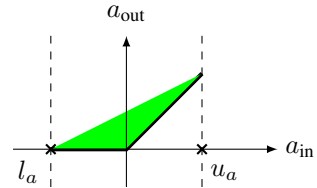
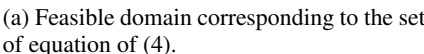
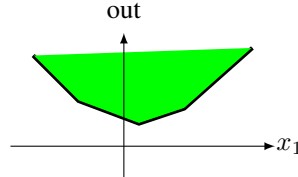

(a) Feasible domain corresponding to the set of equation of (4).

(b) Cut along one dimension of the feasible set of equations (5) where $in_i$ are linear function of the dimension of the cut.

Both Reluplex and Planet rely on the splitting mechanism over ReLUs to guarantee progress but their use of it is fundamentally different: while Reluplex (Katz et al., 2017a) drives the search for satisfiability using a simplex algorithm and splits the ReLU lazily to unlock cases, Planet (Ehlers, 2017a) drives the whole search by eagerly making splits using a SAT solver and making deductions based on those. Reluplex always maintains an assignment to all variables even though it doesn't respect all constraints until the end; Planet only maintains assignments to the phase of the non-linearities. As those two approaches have never been compared on common benchmarks, it is hard to identify which is the most promising one or the specific cases in which one method outperforms the other, even though they rely on similar principles.

### 3.4 BRANCH AND BOUND OPTIMIZATION FOR VERIFICATION

**Verification as optimization** We now present a different way of approaching the Neural Network verification problem. The whole satisfiability problem will be transformed into an optimization problem where the decision will be obtained by checking the sign of the minimum. We will show how any Boolean formula on linear inequalities can be encoded as additional layers at the end of the network.

Assume that the property is a simple inequality: $P(\mathbf{y}) \triangleq \mathbf{c}^T \mathbf{y} \geq b$. In that case, it is sufficient to add to the network a final fully connected layer with one output, with weight of $\mathbf{c}$ and a bias of $-b$. If the global minimum of this network is positive, it indicates that for all $\mathbf{y}$ that the original network can output, we have $\mathbf{c}^T \mathbf{y} - b \geq 0 \implies \mathbf{c}^T \mathbf{y} \geq b$, and as a consequence the property is True. On the other hand, if the global minimum is negative, then it provides a counter-example to the property.

Clauses specified using OR (denoted by $\bigvee$) can be encoded by using a MaxPooling unit; if the property is $P(y) = \bigvee_i \left[ \mathbf{c}_i^T \mathbf{y} \geq b_i \right]$, this is equivalent to $\max_i \left( \mathbf{c}_i^T \mathbf{y} - b_i \right) \geq 0$. Clauses specified using AND (denoted by $\bigwedge$) can be encoded similarly: the property $P(y) = \bigwedge_i \left[ \mathbf{c}_i^T \mathbf{y} \geq b_i \right]$ is equivalent to $\min_i \left( \mathbf{c}_i^T \mathbf{y} - b_i \right) \geq 0 \iff - \left( \max_i \left( -\mathbf{c}_i^T \mathbf{y} + b_i \right) \right) \geq 0$

As a result, we can formulate any Boolean formula over linear inequalities on the output of the network as a sequence of additional layers, and the verification problem would be reduced to a global minimization problem. Aside from some specific class of NN (Amos et al., 2017), this remains a hard problem. The advantage is one of formalism, allowing us to prove complex properties ,containing several OR clauses, with a single procedure rather than having to decompose the desired property into separate queries as was done in previous work (Katz et al., 2017a).

Finding the exact global minimum, while not necessary for verification, will have the advantage of generating a value. If this value is positive, it will correspond to the margin by which the property is satisfied. When estimating robustness to adversarial examples, existing methods choose to perform a binary search over the maximum radius guaranteeing the absence of adversarial examples. The optimization process would be a more appropriate formalism here.

**Branch and Bound for Optimization** Optimization algorithms such as Stochastic Gradient Descent which are the usual workhorses of Deep Learning are not appropriate for this minimization problem. Despite being capable of converging to good local minima, they have no way of guaranteeing whether or not a minima is global. We now present an approach to tackle this problem, based on the Branch-and-Bound method.

Algorithm 1 describes the generic form of the Branch-and-Bound method. The input domain will be repeatedly split into sub-domains (line 7), over which we will compute lower and upper bounds of the minimum of the output (lines 9-10). The best upper-bound found so far will serve as a candidate for the global minimum. Any domain whose lower bound is greater than the current global upper

bound can be pruned away as it necessarily won't contain the global minimum (line 13, lines 15-17). By iteratively splitting the domains, we will be able to compute tighter lower bounds. We keep track of the global lower bound on the minimum by taking the minimum over the lower bounds of all sub-domains (line 19). When the global upper bound and the global lower bound differ by less than a small scalar $\epsilon$ (line 5), we consider that we have converged.

---

**Algorithm 1** Branch and Bound

```
 1: function BAB(net, domain, ε)              ▷ Global minimum of net on domain to an ε accuracy
 2:     global_ub ← inf
 3:     global_lb ← − inf
 4:     domains ← [(global_lb, domain)]
 5:     while global_ub − global_lb > ε do
 6:         (_, dom) ← pick_out(domains)
 7:         [dom_1, ..., dom_s] ← split(dom)
 8:         for i = 1 ... s do
 9:             dom_ub ← compute_upper_bound(net, dom_i)
10:             dom_lb ← compute_lower_bound(net, dom_i)
11:             if dom_ub < global_ub then              ▷ If found a better upper bound
12:                 global_ub ← dom_ub
13:                 domains ← prune_domains(domains, global_ub)  ▷ Remove domains with
                                                                        dom_lb ≥ global_ub
14:             end if
15:             if dom_lb < global_ub then
16:                 domains ← domains + [(dom_lb, dom_i)]
17:             end if
18:         end for
19:         global_lb ← min{lb | (lb, dom) ∈ domains}
20:     end while
21:     return global_ub
22: end function
```

---

The description of the verification problem as optimization and the pseudo-code of Algorithm 1 are generic and would apply to verification problems beyond the specific case of PL-NN. To obtain a practical algorithm, it is necessary to specify the elementary functions:

- `pick_out(domains)`: Select one of the domains to branch on. Several heuristics are possible, based on the bounds previously computed on the domains or based on the size of the domains.

- `split(domain)`: Takes as argument a domain and returns a partition of domains $[\text{dom\_1}, \ldots, \text{dom\_s}]$ such that $\bigcup_i \text{dom\_i} = \text{domain}$ and that $(\text{dom\_i} \cap \text{dom\_j}) = \emptyset,\ \forall i \neq j$. Choosing the split function will define the "shape" of the domains that we are operating on, potentially making the computation of the bounds harder or easier.

- `compute_{lower, upper}_bound(net, domain)`: Compute a (lower / upper) bound of the minimum of net over the feasible domain domain. We want the lower bound to be as high as possible (so that this whole domain can be pruned easily) and the upper bound to be as low as possible (so that we can use it to prune out other regions of the search space). If we have several approaches to compute bounds, we can employ all of them at once and only keep the tightest. In our experiments, we use the result of minimising the variable corresponding to the output of the network, subject to the constraints of the linear approximation introduced by Ehlers (2017a) as lower bound and simple random sampling as upper bound. As the procedure is generic, any improved algorithm leading to faster or tighter bounds would directly translate to improvements to the verification process.

In practice, it is not necessary to run Algorithm 1 to convergence for verification. If a negative global upper bound is found, the corresponding input is a valid counter-example. Similarly, as soon as the global lower bound goes above zero, we know that the property is verified.

Now equipped with an understanding of the workings of each method, we are now ready to compare them.

## 4 SOLVER AND PROPERTIES

### 4.1 DATASET

As each method implements a different strategy and the worst-case analysis always indicate exponential runtimes, these approaches can only be compared experimentally. We attempt to perform verification on three data sets of verification properties and report the comparison results. The dimensions of all of the problems are given in Table 2.

| Data set | Numbers of Properties | Model Architecture |
|---|---|---|
| ACAS | 188 | 5 inputs
6 layers of 50 hidden units
5 outputs |
| CollisionDetection | 500 | 6 inputs
40 hidden unit layer, Maxpool
19 hidden unit layer
2 outputs |
| PCAMNIST | 27 | 10 or {5, 10, 25, 100, 500, 784} inputs
4 or {2, 3, 4, 5, 6, 7} layers
of 25 or {10, 15, 25, 50, 100} hidden units,
1 output, with a margin of +1000 or
{-1e4, -1000, -100, -50, -10, -1 ,1, 10, 50, 100, 1000, 1e4} |

Table 2: Dimensions of all the data sets. For PCAMNIST, we use a base network with 10 inputs, 4 layers of 25 hidden units and a margin of 1000. We generate new problems by changing one hyperparameter at a time, using the values inside the brackets.

### 4.2 METHODS

**Reluplex**, based on the version released by the authors (Katz et al., 2017b). The tool is implemented in C++ and relies on a modified version of the GLPK library to deal with the Simplex algorithm. Note that as the tool doesn't support MaxPooling units out of the box, we automatically convert the MaxPooling layers into a series of linear layers with ReLU activations. To do so, we decompose the element-wise maximum into a series of pairwise maximum

$$\max\left(x_1, x_2, x_3, x_4\right) = \max\left(\ \max\left(x_1, x_2\right),\ \max\left(x_3, x_4\right)\right), \quad (6)$$

and decompose the pairwise maximums as sum of ReLUs:

$$\max\left(x_1, x_2\right) = \max\left(x_1 - x_2,\ 0\right) + \max\left(x_2 - l_{x_2}, 0\right) + l_{x_2}, \quad (7)$$

where $l_{x_2}$ is a pre-computed lower bound of the value that $x_2$ can take. We initially observed some errors due to numerical accuracy appearing on the CollisionDetection dataset which we reported to the authors. They provided us guidance on setting tolerance hyperparameters appropriately, reducing the number of incorrectly classified properties. We report results for this set of parameters.

**Planet**, based on the version released by the author (Ehlers, 2017b). The tool is implemented in C++, using GLPK to solve linear programs and a modified version of MiniSat to drive the search. We wrote some software to convert in both directions between the input format of both Reluplex and Planet. We discovered some memory issues on the original implementation, which we reported to the author and used the fixed version for all experiments.

**MIP**, that consists on encoding the satisfiability problem as a Mixed Integer Program, using the "big M" encoding of the non-linearity. The exact encoding of MaxPooling and ReLU can be found in Appendix A. This method is similar to the one implemented in (Cheng et al., 2017b; Lomuscio & Maganti, 2017) but due to the lack of availability of open-sourced methods, we reimplemented the approach in Python, using the Gurobi MIP solver. To choose the value of $M$ for each ReLU, we made use of the linear approximation of Planet. This leads to better values of M than the dataflow analysis discussed by Cheng et al. (2017b). Even when compared to their other proposed heuristic

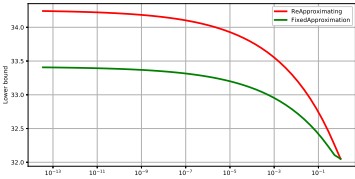
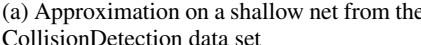
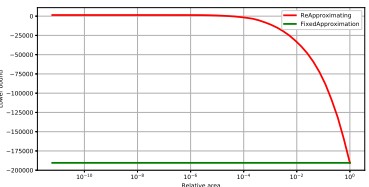

(a) Approximation on a shallow net from the CollisionDetection data set

(b) Approximation on a deep net from the ACAS data set

Figure 3: Quality of the linear approximation, depending on the size of the domain. We compute the value of the lower bound on a given domain, centred around the global minimum and repeatedly shrink the size of the domain. Rebuilding completely the linear approximation at each step allows to create tighter lower-bounds thanks to better $l_a$ and $u_a$, as opposed to using the same constraints and only changing the bounds on input variables. This effect is even more significant on deeper networks.

for the selection of $M$, it has several advantages: it encompasses information from all layers of the network rather than from only a few layers; it solves simple LPs rather than MIPs. The default settings of Gurobi were used. We also include an alternate version, **MIPtune**, where for each dataset, we started by running the Gurobi Tuning tool for a day on one property chosen at random in the dataset so as to pick hyperparameters. Those hyperparameters were then used for all the properties of the dataset.

**BaB**, based on the method described in Section 3.4. Our `pick_out` strategy chooses the domain that currently has the smallest lower bound. We split the domain in half across its longest edge to generate two new smaller domains as our `split` method. We generate upper bounds on the minimum by randomly sampling points on the considered domain, and minimise the linear approximation of the network proposed by (Ehlers, 2017a) as a lower bound. Our implementation is in Python and uses Gurobi to solve LPs. We also include results for a **reluBaB** variant, where the `split` method is based on the phase of the non-linearities, similarly to **Planet** but without using a SAT solver to drive the search. Note that as opposed to the approach taken by Ehlers (2017a) of building a single approximation of the network, we rebuild a new approximation for each sub-domain. This is motivated by the observation shown in Figure 3 which demonstrate the significant improvements it brings, especially for deep networks. In addition, the branch and bounds methods explictly search for upper bounds as well, which has the potential to improve the time at which counterexamples are exhibited.

### 4.3 EVALUATION CRITERIA

For each of the data sets, we compare the different methods using the same protocol. Each run is done with a timeout of two hours, and a maximum allowed memory usage of 20GB, on a single core of a machine with an i7-5930K CPU and 32GB of RAM. We will release all code and data necessary to replicate our analysis.

*Success rate* corresponds to the proportion of properties for which the solver returns the correct answer before timing out or being terminated for using too much memory. We compare the performances of each method and present the results separately for the cases where the properties are True or False. **SAT** means that the satisfiability problem for a counter-example was satisfiable, implying that the property was False. In this case, the runtime corresponds to the time it took before exhibiting a counter-example to the property. On the other hand, **UNSAT** corresponds to the time it took to prove that the problem was infeasible and that no counterexamples to the property could exist. As methods returning **SAT** exhibit a counterexample, disagreement between solvers can be easily resolved by evaluating the network over the counterexample and checking the property on the output. We use this criterion to establish a ground truth for each property. We reported bugs to the original authors of each method every time such a disagreement was detected.

In the case of a timeout, the runtime for the method is counted as the maximum allowed time (7200 seconds), even though the actual runtime would be worse. As a result, the average runtime for methods with low success rate would be worse in practice than reported here. After fixing the Planet solver, no Out of Memory error was encountered. To give more insights into the relative

performance of each solver, we count the *Number of wins*, which corresponds to how many times a solver was the fastest to solve a property. When the relative difference between the runtime of two solvers is less than 1%, we don't count any win.

Having now presented the software tools we use, we now report their performance on each data set.

## 5 ANALYSIS

### 5.1 AIRBORNE COLLISION AVOIDANCE SYSTEM

The Airborne Collision Avoidance System (**ACAS**) data set, as released by Katz et al. (2017a) is a neural network based advisory system recommending horizontal manoeuvres for an aircraft in order to avoid collisions, based on sensor measurements. Each of the five possible manoeuvres is assigned a score by the neural network and the action with the minimum score is chosen. In the case of complex property to prove, such as "NoAction (CoC) is the minimal score", their counter-example search is implemented as a series of four satisfiability problems: "Does there exist a point where the score for {WeakLeft, WeakRight, StrongLeft, StrongRight} is less than the score for NoAction?". While our approach discussed in Section 3.4 would be able to merge all of these satisfiability problems into a single one, we use the original strategy of the authors for a fair comparison.

| Method | Success Rate | Average runtime | | Number of Wins | |
|---|---|---|---|---|---|
| | | **SAT** | **UNSAT** | **SAT** | **UNSAT** |
| Reluplex | 79.26% | 2010.4s | 1775.2 s | 3 | 1 |
| Planet | 46.28% | 7200 s | 2640.3 s | 0 | 0 |
| MIP | 49.47% | 6014.0 s | 2811.84 s | 3 | 1 |
| MIPtune | 46.28% | 6804.0 s | 2732.4 s | 0 | 1 |
| BaB | 83.51% | 128.14 s | 882.76 s | 20 | 91 |
| reluBaB | 85.64% | 157.62 s | 902.57 s | 9 | 35 |

Table 3: Results on the **ACAS** data set. Note that 23 of the properties were not solved by any of the methods and are therefore not present in the SAT/UNSAT breakdowns. Reluplex and BaB/reluBaB boast comparable success rate but the BaBs finish significantly faster, especially for SAT.

Results for the ACAS data are shown in Table 3. Our proposed method using Branch and Bound performs the best along all criteria. Compared to the second best method, Reluplex, it is more than an order of magnitude faster at exhibiting counter-examples and more than twice as fast at proving the correctness of True properties. The variants performing splitting on the non-linearity phase, reluBaB, solves more properties but is on average slightly slower. On this data set where the networks are deep, the lack of updates to the linear approximation make the approximation really poor (see Figure 3), which explains Planet not doing well. The BaB methods are the only ones performing better on **SAT** problems than on **UNSAT** ones. We postulate that this is due to the relatively small dimensionality of the input, which makes random testing capable of easily getting good coverage and discovering counterexamples quickly so the simple upper bound method we employed is good enough to find counterexamples.

The tuning of the MIP solver doesn't improve the performance of the method significantly. Although the performance difference is significant on the property that was used for the tuning (40 times faster), the hyperparameters don't generalize well to other properties; the tuned version being faster than the default one in only half the testcases. This behaviour is also observed for the other datasets.

### 5.2 COLLISIONDETECTION

In the **CollisionDetection** data set, as released by the authors of Planet (Ehlers, 2017a), the network attempts to predict whether two vehicles with parameterized trajectories are going to collide. A total of 500 properties are extracted from problems arising from a binary search to identify the size of the region around training examples in which the prediction of the network doesn't change.

Planet is the best performing method on the set of benchmarks that accompanied the release of the tool, even though it was the worst performing one on ACAS. Table 4 shows it being the fastest method for most of the properties. Conversely, BaB / reluBaB becomes the worst performing

| Method | Success Rate | Average runtime | | Number of Wins | |
|---|---|---|---|---|---|
| | | SAT | UNSAT | SAT | UNSAT |
| Reluplex | 99.8 % | 1.15 s | 1.04 s | 52 | 20 |
| Planet | 100 % | 0.50 s | 0.18 s | 84 | 288 |
| MIP | 100 % | 0.66 s | 0.64 s | 6 | 3 |
| MIPtune | 100 % | 0.64 s | 0.63 s | 17 | 9 |
| BaB | 100 % | 5.27 s | 29.05 s | 0 | 0 |
| reluBaB | 100 % | 23.58 s | 37.06 s | 4 | 0 |

Table 4: Results on the **CollisionDetection** data set. All solvers finished on all test cases but even adjusting the hyperparameters, Reluplex erroneously classified one properties as True. For 17 properties, the difference between the two fastest methods was inferior to 1% of their runtime so we didn't count any Win.

method, especially on True properties. It is however important to note that all solvers finished significantly below the timeout limit, indicating that the data set isn't extremely challenging.

On this data set, Reluplex classified one False properties as unsatisfiable. We evaluated the counterexamples returned by other solvers using Reluplex's code and confirmed that they were valid counterexamples. Using other settings of tolerance parameters, this property would be appropriately classified but these settings lead to other properties being misjudged.

## 5.3 PCAMNIST

To get a better understanding of what factors influence the performance of various methods, we propose a novel **PCAMNIST** data set, which we generated to have control over different hyperparameters. The networks takes $k$ features as input, corresponding to the first $k$ eigenvectors of a Principal Component Analysis of the digits from the MNIST data set. We train the network to predict whether the represented numbers are odd or even. Obtained accuracies are given in Appendix B.

For each network, we attempt to prove whether or not there exists an input for which the score assigned to the odd class is greater than the score of the even class plus a large confidence. We heuristically determine what the largest such confidence can be by running gradient descent with random restarts for a long period of time. Knowing this maximum confidence, we can choose the value of the confidence differential we want to prove so as to have the property be true (or false) by a margin of our choosing.

Table 5 gives the overall results over the data set but given that all properties have different parameters, global summaries may not be an adequate representation. It is however notable that Reluplex which was one of the best performing method on ACAS is here one of the worst. We also present the result in Figure 4, where we generate a plot showing the evolution of runtimes for each of the architectural parameters of the networks. This allows us to assess the impact of each factor on the performance of the solvers.

For all graphs of Figure 4, the trend for all the methods are similar, which seems to indicate that hard properties are intrinsically hard and not just hard for a specific solver. This doesn't mean that

| Method | Success Rate | Average runtime | | Number of Wins | |
|---|---|---|---|---|---|
| | | SAT | UNSAT | SAT | UNSAT |
| Reluplex | 18.52 % | 4800.8 s | 6171.5 s | 0 | 0 |
| Planet | 33.33 % | 7200 s | 4119.7 s | 0 | 3 |
| MIP | 51.85 % | 4800.8 s | 3091.7 s | 2 | 6 |
| MIPtune | 51.85 % | 4801.0 s | 3091.5 s | 0 | 2 |
| BaB | 55.56 % | 2403.1 s | 3448.1 s | 1 | 0 |
| reluBaB | 48.15 % | 2407.3 s | 4220.4 s | 1 | 0 |

Table 5: Results on the **PCAMNIST** data set. 11 properties were not solved by any of the methods. Branch and Bounds methods are on average faster for the case where properties are False and only a counterexample needs to be exhibited. In the case where the property is True, MIP is the best performing method.

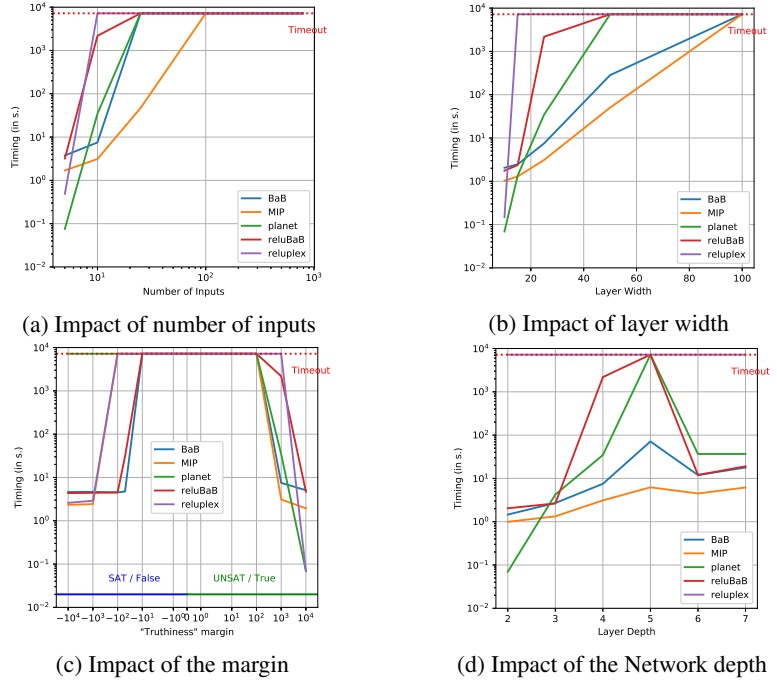

(a) Impact of number of inputs

(b) Impact of layer width

(c) Impact of the margin

(d) Impact of the Network depth

Figure 4: Impact of the various hyperparameters over the runtimes of the different solvers. The base network has 10 inputs and 4 layers of 25 hidden units, and the property to prove is True with a margin of 1000. Each of the subgraph correspond to a variation of one of this parameters. MIPtune is omitted as the curves was almost identical to MIP.

all solvers have identical performance profiles. For example, in the easiest settings where all the solvers are succesful at solving a property, Planet is always the fastest solver, indicating that in case of easy problems, it is the most appropriate. On the other hand, MIP can be considered the more robust solver as it performs the best on hard tasks when most of the other solvers fail.

Figure 4a shows an expected trend: the largest the number of inputs, the harder the problem is. Similarly, Figure 4b shows unsurprisingly that wider networks require more time to solve, which can be explained by the fact that they have more non-linearities. The impact of the margin, as shown in Figure 4c is also clear: properties that are clearly True or clearly False are easiest to prove, while properties that are on the limit are significantly harder. Note that as opposed to the other solvers, Planet has a lot more difficulties to find counterexamples, even when the property is False by a significant margin. Figure 4d presents some intriguing results: after a certain point, increasing the depth of the network makes the problem easier. Note however than this is only considering the impact of the network depth once we have fixed the value of the margin and not the impact of the network depth by itself.

## 6 CONCLUSION

The improvement of formal verification of Neural Networks represents an important challenge to be tackled before learned models can be used in safety critical applications. The lack of a shared set of benchmarks between researchers makes it hard to evaluate progress or estimate promising research directions. We gathered test cases from existing literature, proposed new ones, and evaluated the performance of published methods, which allowed us to surface issues in published tools and offer an informed view of the status of the field. While no clear winner emerged from all the data sets, the importance of testing methods on various datasets appears clearly from the fact that an approach performing the best on its dataset of choice can also be the worst performing on another one so any improvement claimed should be demonstrated on a variety of tasks.

In addition, we proposed a conceptually simple method that offered competitive performance on the challenging data sets. Our method is sufficiently general to be easily improvable, as any better lower bound will directly translate to faster verification.

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

## A MIP ENCODING DETAILS

We now present the encoding used to convert the non linearity of a neural network into inequality constraints involving binary variables.

In the case of a ReLU activation

$$a_{\text{out}} = \max\left(a_{\text{in}}, 0\right) \tag{8}$$

we can replace this equation containing the $\max$ non-linearity by

$$
\begin{array}{cc}
a_{\text{out}} \geq 0 & a_{\text{out}} \geq a_{\text{in}} \\
a_{\text{out}} \leq a_{\text{in}} + (1 - \delta_a)M_a & a_{\text{out}} \leq \delta_a M_a \\
\multicolumn{2}{c}{\delta_a \in \{0, 1\}}
\end{array}
\tag{9}
$$

where $M_a$ is a value such that $M_a$ is an upper-bound of $a_{\text{in}}$ and $-M_a$ is a lower-bound of $a_{\text{in}}$.

For MaxPooling units, we can either make the choice to decompose the MaxPooling into a series of ReLU and use their linear encoding. Another solution is to encode the MaxPooling directly. The constraint

$$\text{out} = \max_i\left(\text{in}_i\right) \tag{10}$$

can be replaced by

$$
\begin{aligned}
& \text{out} \geq \text{in}_i \quad \forall i \\
& \text{out} \leq \text{in}_i + (M - \text{lb}_i)(1 - \delta_i) \quad \forall i \\
& \sum_i \delta_i = 1 \\
& \delta_i \in \{0, 1\} \quad \forall i
\end{aligned}
\tag{11}
$$

where $M$ is an upper-bound on all $\text{in}_i$ and $\text{lb}_i$ are lower-bounds on each $\text{in}_i$.

## B PCAMNIST ACCURACIES

| Network Parameter | | | Accuracy | |
|---|---|---|---|---|
| Nb inputs | Width | Depth | Train | Test |
| 5 | 25 | 4 | 88.18% | 87.3% |
| 10 | 25 | 4 | 97.42% | 96.09% |
| 25 | 25 | 4 | 99.87% | 98.69% |
| 100 | 25 | 4 | 100% | 98.77% |
| 500 | 25 | 4 | 100% | 98.84% |
| 784 | 25 | 4 | 100% | 98.64% |
| 10 | 10 | 4 | 96.34% | 95.75% |
| 10 | 15 | 4 | 96.31% | 95.81% |
| 10 | 25 | 4 | 97.42% | 96.09% |
| 10 | 50 | 4 | 97.35% | 96.0% |
| 10 | 100 | 4 | 97.72% | 95.75% |
| 10 | 25 | 2 | 96.45% | 95.71% |
| 10 | 25 | 3 | 96.98% | 96.05% |
| 10 | 25 | 4 | 97.42% | 96.09% |
| 10 | 25 | 5 | 96.78% | 95.9% |
| 10 | 25 | 6 | 95.48% | 95.2% |
| 10 | 25 | 7 | 96.81% | 96.07% |

Table 6: Accuracies of the network trained for the PCAMNIST dataset.

