# OpenReview forum: "Piecewise Linear Neural Networks verification: A comparative study"
_ICLR.cc/2018/Conference — Reject_

### Official Review · AnonReviewer3 · 2017-11-20
**Interesting study with some flaws**

**Rating:** 6
**Confidence:** 3

**Review:**

The paper studies methods for verifying neural nets through their piecewise
linear structure. The authors survey different methods from the literature,
propose a novel one, and evaluate them on a set of benchmarks.

A major drawback of the evaluation of the different approaches is that
everything was used with its default parameters. It is very unlikely that these
defaults are optimal across the different benchmarks. To get a better impression
of what approaches perform well, their parameters should be tuned to the
particular benchmark. This may significantly change the conclusions drawn from
the experiments.

Figures 4-7 are hard to interpret and do not convey a clear message. There is no
clear trend in many of them and a lot of noise. It may be better to relate the
structure of the network to other measures of the hardness of a problem, e.g.
the phase transition. Again parameter tuning would potentially change all of
these figures significantly, as would e.g. a change in hardware. Given the kind
of general trend the authors seem to want to show here, I feel that a more
theoretic measure of problem hardness would be more appropriate here.

The authors say of the proposed TwinStream dataset that it "may not be
representative of real use-cases". It seems odd to propose something that is
entirely artificial.

The description of the empirical setup could be more detailed. Are the
properties that are being verified different properties, or the same property on
different networks?

The tables look ugly. It seems that the header "data set" should be "approach"
or something similar.

In summary, I feel that while there are some issues with the paper, it presents
interesting results and can be accepted.

---

> ### Author Response · Authors · 2017-12-08
> **About Hyperparameters and Empirical comparison - Reply to Reviewer 3**
>
> We thank the reviewer for reading our paper and providing comments to improve our paper.
>
> Hyperparameters:
> The reviewer mentions drawbacks in the evaluation of the paper due to the lack of hyperparameter tuning for each benchmark.
> Planet has very little: going through their source code we found AGGREGATED_CHANGE_LIMIT used at initialization of their linear approximation and an EPSILON(...) parameter for deciding that inequalites are strict. While we didn’t explicitly experiment with the AGGREGATED_CHANGE_LIMIT parameter of Planet, experimenting with doing repeated optimization of nodes without branching (as this hyperparameters controls) didn’t significantly improve. We will observe the impact of this hyperparameter on some example of the ACAS dataset and rerun the experiment if a significant difference is observed.
> Reluplex has more, mostly dedicated to addressing numerical errors. Since the submission, we had some input from the authors of Reluplex to address the numerical errors observed on TwinStream and CollisionDetection and re-ran experiments with updated parameters under their recommandation. This lead to minor speed improvements, which however didn’t change the story told by the results presented on each dataset. All those new results will be uploaded in the next version of the paper.
> MIP might have more hyperparameters if we consider all the hyperparameters that a solver like Gurobi gives access to (http://www.gurobi.com/documentation/7.5/refman/parameters.html ) . We indeed did not tune those parameters specifically, which might result in performance variance. Given the already extremely long runtime of verification experiments, it is hard to perform exhaustive tuning over this large space of parameters. We will use the gurobi tuning tool (http://www.gurobi.com/documentation/7.5/refman/tuning_api.html) to find a set of hyperparameters appropriate for each dataset based on a few properties and rerun the experiments using these.
> The branch-and-bound strategy doesn’t have itself any parameters, unless you consider the choice of elementary functions (pick_out, split, compute_lower_bounds, compute_upper_bounds) hyperparameters.
> We would appreciate any additional comments on hyperparameters needing tuning that we might not be aware of.
>
> Regarding the artificiality of TwinStream:
> As mentioned in the general comment, it was useful in helping us diagnose numerical accuracy problems of networks and the importance of reapproximation for deeper network but we agree that it might be limited to a toy-problem useful for developing methods and would not be useful to draw larger conclusions from it. We will run additional experiments on a new benchmark composed of Networks of various architecture / margins but based on a real learning task and replace it.
>
> Regarding a more theoretic measure of problem hardness to better show differences.
> It seems to me that given that the verification problem is NP-hard, any informative measure of hardness would necessarily be based on experimental results. Piecewise Neural Network verification being a problem that only recently started being studied, I’m not aware of any measure of theoretical hardness for specific problems.
>
> Regarding the description of the empirical setups.
> We will add more details to the paper. To answer the reviewers question, the ACAS dataset has 45 different networks and several properties are defined over each network. The appendix of https://arxiv.org/pdf/1702.01135.pdf contains the exact list of network - properties it contains. The CollisionDetection dataset is a set of 500 properties on the same network. In TwinStream, each property is a unique network.
>
> We will improve the look of the tables and correct the wrong headers in the next submitted version.

---

> > ### Comment · AnonReviewer3 · 2017-12-08
> > **reply to reply**
> >
> > Thank you for your reply.
> >
> > Regarding hyperparameters: anything that makes sense to be changed should be tuned. It does not need to be exhaustive tuning, and there is software available to do this (e.g. spearmint, irace, smac). It might not make a difference in many cases, but especially for MIP solvers parameter tuning can cause massive differences.
> >
> > Regarding the measure of hardness: what about the phase transition? It seems like this would be a more robust measure.

---

> > > ### Author Response · Authors · 2017-12-12
> > > **Tuning Tool - Phase Transition discussion**
> > >
> > > We will perform proper tuning of the hyperparameters of the MIP solver using the Gurobi tuning tool specially adapted to our MIP solver and include those results.
> > >
> > >
> > > Regarding the phase transition, we assume that the reviewer is referring to the phase transition in satisfiability problems: Depending on the ratio of clauses to variables, for problem such as 3-SAT, the result go from most likely SAT to most likely unsat, with a phase transition in between those two regimes containing the hardest instances. We apologize if we misunderstood the point that the reviewer was making.
> > >
> > > If this is what was suggested, it’s not certain that NN verification exhibits such a behaviour with regards to parameters such as depth / number of hidden units or number of inputs. It however seems possible that this would be the case with regards to the margin of the property to prove.
> > > If the property to prove is True with a very high margin, this is going to be an easy proof as any branch-and-bound type method won’t need to do a lot of branching. If the property is False with a very high margin as well, it will be probably be easy to exhibit a counterexample as they would be many. On the other hand, if the margin is close to zero, very few counter-examples might exist (if the property is false -> margin is negative), making them hard to find or getting tight enough bounds will require a lot of branching (if the property is True -> margin is positive). We already see some evidence of this in Figure 7: making the margin go towards zero makes the problems harder, for all of the solvers. We will add some experiments with negative margins to see if the same conclusion can be drawn on the other side of the limit point / phase transition.
> > >
> > >
> > > It’s not obvious however how general this measure of hardness will be. Multiplying all weights and biases of the last layer by 10 will similarly scale the margin, without significantly changing the hardness of the problem.

---

### Official Review · AnonReviewer1 · 2017-11-23

**Rating:** 5
**Confidence:** 4

**Review:**

Summary:

This paper:
- provides a compehensive review of existing techniques for verifying properties of neural networks
- introduces a simple branch-and-bound approach
- provides fairly extensive experimental comparison of their method and 3 others (Reluplex, Planet, MIP) on 2 existing benchmarks and a new synthetic one

Relevance: Although there isn't any learning going on, the paper is relevant to the conference.

Clarity: Writing is excellent, the content is well presented and the paper is enjoyable read.

Soundness: As far as I can tell, the work is sound.

Novelty: This is in my opinion the weakest point of the paper. There isn't really much novelty in the work. The branch&bound method is fairly standard, two benchmarks were already existing and the third one is synthetic with weights that are not even trained (so not clear how relevant it is). The main novel result is the experimental comparison, which does indeed show some surprising results (like the fact that BaB works so well).

Significance: There is some value in the experimental results, and it's great to see you were able to find bugs in existing methods. Unfortunately, there isn't much insight to be gained from them. I couldn't see any emerging trend/useful recommendations (like "if your problem looks like X, then use algorithm B"). This is unfortunately often the case when dealing with combinatorial search/optimization.

---

> ### Author Response · Authors · 2017-12-08
> **About datasets and insight from experiments - Reply to Reviewer 1**
>
> We thank the reviewer for the comments. We agree that the main novel contribution of our paper lies in the experimental comparison between various methods which was lacking from the literature. Branch&Bound is indeed a fairly standard method when a global minimum is required but we would like to point out that some insights were also brought up in the way lower bounds are computed (namely the importance of rebuilding the approximation at each step, Figure 3).
>
> Regarding surfacing trends, some conclusion can already be made. The CollisionDetection dataset seems to be easily solvable by every method, which might limit the significance of its result. ACAS represents more of a significant challenge (especially given that in the alloted time, certain properties aren’t solved by any methods) and might be where one draws concrete conclusions. The additional TwinStream benchmark would hopefully have confirmed these observations but it’s not clear whether the drastically different results are due to a lack of trends or simply to the artificialness of the benchmark revealing itself. While it was useful in helping us diagnose numerical accuracy problems of networks and the importance of reapproximation for deeper network, that might be the extent of its usefulness. We will run additional experiments on a new benchmark composed of Networks of various architecture / margins but based on a real learning task and replace it.

---

### Official Review · AnonReviewer2 · 2017-11-26
**no original contribution**

**Rating:** 3
**Confidence:** 5

**Review:**

The paper compares some recently proposed method for validation of properties
of piece-wise linear neural networks and claims to propose a novel method for
the same. Unfortunately, the proposed "branch and bound method" does not explain
how to implement the "bound" part ("compute lower bound") -- and has been used
several times in the same application, incl.:

Ruediger Ehlers. Planet. https://github.com/progirep/planet,
Chih-Hong Cheng, Georg Nuhrenberg, and Harald Ruess.  Maximum resilience of artificial neural networks. Automated Technology for Verification and Analysis
Alessio Lomuscio and Lalit Maganti.  An approach to reachability analysis for feed-forward relu neural networks. arXiv:1706.07351

Specifically, the authors say: "In our experiments, we use the result of
minimising the variable corresponding to the output of the network, subject
to the constraints of the linear approximation introduced by Ehlers (2017a)"
which sounds a bit like using linear programming relaxations, which is what
the approaches using branch and bound cited above use. If that is the case,
the paper does not have any original contribution. If that is not the case,
the authors may have some contribution to make, but have not made it in this
paper, as it does not explain the lower bound computation other than the one
based on LPs.

Generally, I find a jarring mis-fit between the motivation (deep learning
for driving, presumably involving millions or billions of parameters) and
the actual reach of the methods proposed (hundreds of parameters).
This reach is NOT inherent in integer programming, per se. Modern solvers
routinely solve instances with tens of millions of non-zeros in the constraint
matrix, but require a strong relaxation. The authors may hence consider
improving the LP relaxation, noting that the big-M constraint are notorious
for producing weak relaxations.

---

> ### Public Comment · (anonymous) · 2017-12-03
> **Trying to parse the review**
>
> "Generally, I find a jarring mis-fit between the motivation (deep learning
> for driving, presumably involving millions or billions of parameters) and
> the actual reach of the methods proposed (hundreds of parameters).
> This reach is NOT inherent in integer programming, per se. Modern solvers
> routinely solve instances with tens of millions of non-zeros in the constraint
> matrix, but require a strong relaxation. The authors may hence consider
> improving the LP relaxation, noting that the big-M constraint are notorious
> for producing weak relaxations."
>
> --  Not questioning the review, but does the reviewer feel SMT or MILP based approaches to verification are not meaningful? I ask this because of the phrase 'jarring misfit' between the goal of deep learning for driving and the 'reach of the methods proposed '.
>
>  -- It is not the number of parameters, rather the number of integer choices that make it hard. I do understand when the reviewer suggests that the bounds generated this way are generally loose (https://arxiv.org/pdf/1711.00851.pdf), and that might be a part of the problem. LPs with 10^9 parameters are easily handled these days.
>
> -- The disjunctions make life difficult for SMT solvers.
>
> -- The title reads 'comparative study'. I think the authors make it clear what the contribution might be.
>
> -- The architecture for ACAS for example has ~13k parameters, and 300 nodes. It's a decent sized network. The field of verifying deep nets is a few years old and in its infancy. If the contributions in the field can push it another 2-3 orders of magnitude, we might see it reach verifying 'deep learning for driving' problems
>
> PS. I have nothing to do with the authors, but a random bystander trying to identify possible directions to make contributions. I haven't even fully read the paper, just confused as to what this review is trying to convey.

---

> ### Author Response · Authors · 2017-12-08
> **Contribution of the Paper - Reply to Reviewer 2**
>
> Thank you for reading our paper and providing comments. Current state of the art methods are at the moment limited to small / medium sized networks that indeed are not representative of the largest networks used in, for example, computer vision. Our dataset of properties reflects this reality. One precondition if we want to make progress and hope to one day be able to verify larger models (such as the one used in autonomous driving) is to be able to accurately assess the strengths and weaknesses of different methods. Benchmarks such as ours are a necessary step for this.
> By establishing runtimes on a common set of test cases, and releasing the code to do so (including adapters to account for models with different capabilities), we hoped to highlight promising directions and facilitate research for others. Even as of now, our comparison already proved useful in discovering bugs in released implementations of previous methods.
>
> We wish to highlight once again that the main contribution of our paper is in the experimental comparison and not the branch and bound method we described. The reviewer is correct in understanding that in our experiment, we used the linear programming relaxation of (Ehlers, 2017) (which is however not exactly the relaxation obtained by dropping the integrality constraint in the big-M IP formulation). Note that our use of this relaxation is different than the one made by Planet: rather than building a single linear approximation at the beginning, we rebuild it after each branching step, and shows that it makes a significant difference (see Figure 3), which might explain in a large amount the performance difference between Planet and BaB on the large networks.
>
> Please note that branch-and-bound as a method has not been used several times in the same application. Lomuscio & Maganti (2017) and Cheng et al. (2017b) simply formulates the problem as a MIP and rely on black-box solvers (respectively Gurobi and CPlex). While these solvers might employ branch-and-bound internally, it is not clear what bounding method they used. Furthermore, they are limited by the intermediate big-M formulation, which the reviewer accurately pointed out to be notorious for producing weak relaxations. The large difference in performance observed between these methods and our BaB also clearly seem to indicate that they are not operating in the same manner.
> Planet is motivated as an SMT solver, which we agree can be cast as a specific form of branch and bound, where branching is restricted to be over decision variables (phase of the hidden units of the network). Note however that under its formulation, Planet can only be used for satisfiability problem and will not be able to give any information over the margin by which a property is true (unless a costly binary search is employed), while a BaB approach can return such an information (provided no early stopping is employed).

---

### Author Response · Authors · 2017-12-08
**To all reviewers**

We thank all the reviewers and the anonymous commenters for reading our paper and giving us valuable feedback.

The main comment is with regards to the lack of novelty. While we introduced a method based on the branch-and-bound framework that surprisingly performed better than other methods on the challenging ACAS dataset, the main contribution is the gathering of existing benchmarks into a common format and the establishment of an experimental comparison.

We take into account the shared criticism that the additional dataset we introduced with the intent of offering insights into the influence of various parameters might be too artificial. While it was useful for us to discover numerical instability problems and made us discover the impact of depth on the importance of improving approximation, the fact that it isn’t representative of real, trained network limits its applicability and potentially explains why no clear trend could be drawn. We will replace it with a more interesting dataset based on networks trained on a shared task, while still varying these parameters.

We will also address specific comments as replies to each review.

---

### Decision · Program_Chairs · 2018-01-29
**ICLR 2018 Conference Acceptance Decision**

**Decision:**

Reject

**Comment:**

All three reviewers are in agreement that this paper is not ready for ICLR in its current state. Given the pros/cons, the committee feels the paper is not ready for acceptance in its current form.